# Antenatal corticosteroids for impending late preterm (34-36+6 weeks) deliveries—A systematic review and meta-analysis of RCTs

Mangesh Deshmukh[1,2,3☉¤a]*, Sanjay Patole[3,4☉¤b]*

1 Department of Neonatalogy, Fiona Stanley Hospital, Perth, Western Australia, 2 Department of Neonatalogy, St. John of God Hospital, Subiaco, Perth, Western Australia, 3 School of Medicine, University of Western Australia, Perth, Western Australia, 4 Department of Neonatal Pediatrics, King Edward Memorial Hospital, Perth, Western Australia

☉ These authors contributed equally to this work.
¤a Current address: Department of Neonatology, Fiona Stanley Hospital, Perth, Western Australia
¤b Current address: Neonatal Directorate, King Edward Memorial Hospital, Perth, Western Australia
* Mangesh.Deshmukh@health.gov.au (MD); Sanjay.patole@health.wa.gov.au (SP)

**Data Availability Statement:** All relevant data are within the paper and its Supporting information files.

## Abstract

### Background

Administration of antenatal corticosteroids (ANC) for impending preterm delivery beyond 34 weeks of gestation continues to be a controversial issue despite various guidelines for obstetricians and gynaecologists.

### Objective

To compare outcomes following exposure to ANC for infants born between 34–36$^{+6}$ weeks' gestation.

### Methods

A systematic review of randomised controlled trials (RCT) reporting neonatal outcomes after ANC exposure between 34–36+6 weeks' gestation using Cochrane methodology. Databases including PubMed, Embase, Emcare, Cochrane Central library and Google Scholar were searched in May 2020. Primary outcomes: (1) Need for respiratory support (Mechanical ventilation, CPAP, high flow) or oxygen (2) Hypoglycemia. Secondary outcomes included respiratory distress syndrome (RDS), transient tachypnoea of newborn (TTN), need for neonatal resuscitation at birth [only in the delivery room immediately after birth (not in neonatal intensive care unit (NICU)], admission to NICU, mortality and developmental follow up. Level of evidence (LOE) was summarised by GRADE guidelines.

### Main results

Seven RCTs (N = 4144) with low to high risk of bias were included. Only one RCT was from high income countries, Meta-analysis (random-effects model) showed (1) reduced need for respiratory support [5 RCTs (N = 3844); RR = 0.68 (0.47–0.98), p = 0.04; I$^2$ = 55%; LOE: Moderate] and (2) higher risk of neonatal hypoglycaemia [4 RCTs (N = 3604); RR = 1.61

**Funding:** The authors received no specific funding for this work.

**Competing interests:** The authors have declared that no competing interests exist.

(1.38–1.87), p<0.00001; $I^2$ = 0%; LOE: High] after ANC exposure. Neonates exposed to ANC had reduced need for resuscitation at birth. The incidence of RDS, TTN and surfactant therapy did not differ significantly. None of the included studies reported long-term developmental follow up.

## Conclusions

Moderate quality evidence indicates that ANC exposure reduced need for respiratory support, and increased the risk of hypoglycaemia in late preterm neonates. Large definitive trials with adequate follow up for neurodevelopmental outcomes are required to assess benefits and risks of ANC in this population.

## Introduction

Administration of antenatal corticosteroids (ANC) is standard practice for threatened preterm delivery between 24–34 weeks' gestation. Based on the evidence from randomised controlled trials (RCTs), most of the Obstetric and Gynaecological College/Society guidelines recommend ANC between 24–34 weeks of gestation [1–3]. However administration of ANC beyond 34 weeks of gestation continues to be a controversial issue, more so for late preterm gestations from 34 to $36^{+6}$ weeks.

Late preterm neonates (LPNs) represent ~70% of total preterm births, which account for ~10% of total births [4]. Compared to neonates born at term (37–40 weeks' gestation), LPNs are at risk of complications such as the need for resuscitation at birth, respiratory distress, hypothermia, and hypoglycaemia. Considering the size of their population, LPNs impose an enormous burden on the health system [5]. A population-based prospective study from UK (N = 1146) found that LPNs are more likely to require resuscitation (17.5% vs 7.4%), respiratory (11.8% vs 0.9%) and nutritional support (3.5% vs 0.3%) and less likely to be fed breast milk (64.2% vs 72.2%) compared to term neonates [6]. Furthermore, LPNs are at six to sevenfold higher risk of complications including transient tachypnoea of newborn (TTN) and respiratory distress syndrome (RDS) compared to term neonates [7, 8]. Admission to the neonatal intensive care unit (NICU) for such complications inevitably means separating the mother-infant dyad with the risk of lactation failure and increased parental anxiety. The average duration of hospital stay has been reported to be longer (8.8 vs. 2.2 days), accounting for 10-fold higher cost of care in LPNs vs. term neonates [9]. Overall, the importance of reducing the enormous health burden especially due to respiratory complications in LPNs cannot be overemphasised.

Based on the results of the Antenatal Late Preterm Steroids (ALPS) study, the American College of Obstetricians and Gynaecologists' (ACOG) guidelines recommend ANC between $34–36^{+6}$ weeks of gestation for reducing neonatal respiratory morbidity, particularly TTN [2, 10]. A recent systematic review and meta-analysis including 3 RCTs (n = 3200) found that ANC reduced the risk of TTN (RR: 0.72, 95% CI: 0.56 to 0.92), severe RDS (RR: 0.60, CI: 0.33–0.94), and need for surfactant (RR: 0.61CI: 0.38 to 0.99) in LPNs. However, the risk of hypoglycaemia was significantly higher (RR: 1.61, CI: 1.38–1.87) in neonates exposed to ANC [11]. Data on these outcomes was available from only 2 of the 3 included RCTs with the ALPS study accounting for 2800 of the 3200 neonates included in this meta-analysis [10]. Importantly, none of these trials reported neonatal long-term neurodevelopmental outcomes.

Considering the difficulties in balancing short-term gains vs. potentially serious long-term adverse effects there is no international consensus for prophylactic ANC at 34 to 36 weeks' gestation despite the ACOG recommendations [12, 13]. Given the clinical significance of this issue, we aimed to conduct a systematic review on the effects of ANC on LPNs.

## Materials and methods

The Cochrane methodology and Preferred Reporting Items for Systematic Reviews and Meta-Analyses (PRISMA) guidelines were used to conduct and report this systematic review respectively [14, 15]. Ethics approval was not required. We have not registered the protocol with PROSPERO or any other database.

### Participants

**Inclusion criteria.** Neonates born between (34–36$^{+6}$) weeks of gestation.

**Exclusion criteria.** Major chromosomal and congenital anomalies.

**Intervention.** Antenatal glucocorticosteroids of any type (e.g. Betamethasone, Dexamethasone), dose (single/multiple), and duration vs. placebo/control.

**Outcomes.** *Primary*. (1) Need for any respiratory support (Mechanical ventilation, CPAP, high flow) or oxygen (2) Hypoglycaemia: Blood glucose level <2.6 mmol/l or as defined by the authors of included studies.

*Secondary*. (1) Need for resuscitation at birth: Requirement of any intervention including positive pressure ventilation, CPAP, facial oxygen (defined as the free flow of oxygen near the nostrils by a catheter or mask) only in the delivery room immediately after birth (not in NICU) (2) Admission to NICU (3) TTN: Tachypnoea, chest x-ray showing increased perihilar intestinal marking or fluid in the fissure. (4) RDS: Clinical signs of respiratory distress such as tachypnoea, rib recessions, grunt, requirement of oxygen, with reticulogranular pattern on chest x-ray (5) Mortality: Death before discharge from the NICU during the first admission after birth (6) Need for mechanical ventilation (7) Need for surfactant: (8) Developmental follow up outcomes (9) Adverse effects including sepsis, and seizures.

**Search strategy.** We searched MEDLINE (from 1966), EMBASE (from1980), CINAHL and Cochrane Central Register of Controlled Trials initially in Dec 2019 and May 2020 for published studies. We used the following search terms in various combinations: a) Population: Neonate(s), newborn(s), infant*, premature, late preterm b) Intervention: Antenatal corticosteroids, adrenocortical stimulating hormone, Betamethasone, Celestone, Dexamethasone, c) Publication type: "Randomized controlled Trial, "Controlled Trial", or "Clinical Trial". Online abstracts of Pediatric Academic Society (PAS) meetings were reviewed from 2002. Abstracts of conference proceedings including Perinatal Society of Australia and New Zealand (PSANZ), European Academy of Paediatric Societies, and the British Maternal and Fetal Medicine Society were searched in EMBASE. We searched 'Google Scholar' for articles that might not have been cited in the standard medical databases. The reference lists of identified studies and reviews were searched to identify additional eligible studies. We also searched www.clinicaltrials.gov and Australian New Zealand trial registry (www.anzctr.org.au) for ongoing studies. No language restriction was applied. Reviewers MD, SP and RM (librarian) conducted the literature search independently.

**Study selection.** Both reviewers (MD and SP) independently selected studies for inclusion in the review. First, the records were screened according to the titles and abstracts. Full texts of the selected articles were then retrieved and assessed for inclusion according to the pre-specified selection criteria.

**Data extraction and management.** Reviewers MD and SP extracted the data independently, using a data collection form. We included the information about authors, year of publication, the country where the study was conducted, setting, inclusion and exclusion criteria, participants characteristics, type of steroids (betamethasone or dexamethasone) used, outcome measures (need for respiratory support, hypoglycaemia, RDS, TTN, neonatal resuscitation, admission to neonatal unit, mortality and developmental follow up) and their definitions. We checked the number of participants allocated to each arm, methods of analysis, loss to follow up and reasons for the same. For dichotomous outcomes, the number of patients with the event and the number of patients analysed in each treatment group was recorded. For continuous outcomes, we recorded the mean and standard deviations. Both reviewers verified the information about study design and outcomes. Discrepancies were resolved by discussion and consensus.

**Assessment of Risk of Bias (ROB).** Both reviewers (MD and SP) assessed the ROB in each included trial for the following seven components: random sequence generation, allocation concealment, blinding of participants and personnel, blinding of outcome assessment, incomplete outcome data, selective outcome reporting and other biases. For each of these components, they assigned ratings of high, low or unclear ROB [16]. Differences in judgements were resolved by discussion.

**Data synthesis.** Meta-analysis was conducted using Review Manager 5.4 [Cochrane Collaboration, Nordic Cochrane Centre], with 'intention to treat analysis'. Heterogeneity between trials was assessed by visual examination of the forest plot to check for overlapping of confidence intervals (CI), Chi$^2$ test and I$^2$ statistics. We used random-effects model (REM) assuming high heterogeneity. However, results were also compared using fixed-effect model (FEM). Categorical and continuous measures of effect size were expressed as risk difference (RR) (Mantel Haenszel method) and mean difference (MD) (Inverse Variance method) respectively. Sensitivity analysis was planned for studies with low ROB. Subgroup analyses were planned by neonatal gestation at birth (34, 35, 36 weeks), mode of delivery (vaginal or caesarean section) and presence of maternal gestational diabetes.

**Assessment of publication bias.** We planned to assess publication bias using a funnel plot [17].

**Grading the evidence and summary of findings.** We used the Grading of Recommendations Assessment, Development and Evaluation (GRADE) for assessment and Grade pro guidelines development tool to create the summary of findings table for reporting the level of evidence (LOE) [18, 19]. Evidence from RCTs was considered as high quality. We graded the evidence in the following domains: ROB, inconsistency, indirectness, imprecision and publication bias. The evidence was downgraded one level for serious and two levels for very serious limitation.

## Results

The literature search retrieved 830 potentially relevant citations (Fig 1). Total of 203 citations removed as duplicate. After carefully reviewing the abstracts and titles 598 citations were excluded. Total 29 citations were read in details and 22 were excluded for reasons mentioned in the flow chart. Finally, 7 RCTs including 4144 neonates whose mothers received ANC between 34–36$^{+6}$ weeks' gestation were eligible for inclusion in the systematic review [10, 20–25]. The type, dose and duration of ANC prophylaxis varied in these 7 RCTs (Dexamethasone: 4, Betamethasone: 3). Five studies provided data for the primary outcome of need for any respiratory support [10, 20–22, 25] whereas only 4 reported on hypoglycemia [10, 20, 21, 25]. The ALPS study by Gyamfi-Bannerman provided data for all outcomes included in this review [10]. The

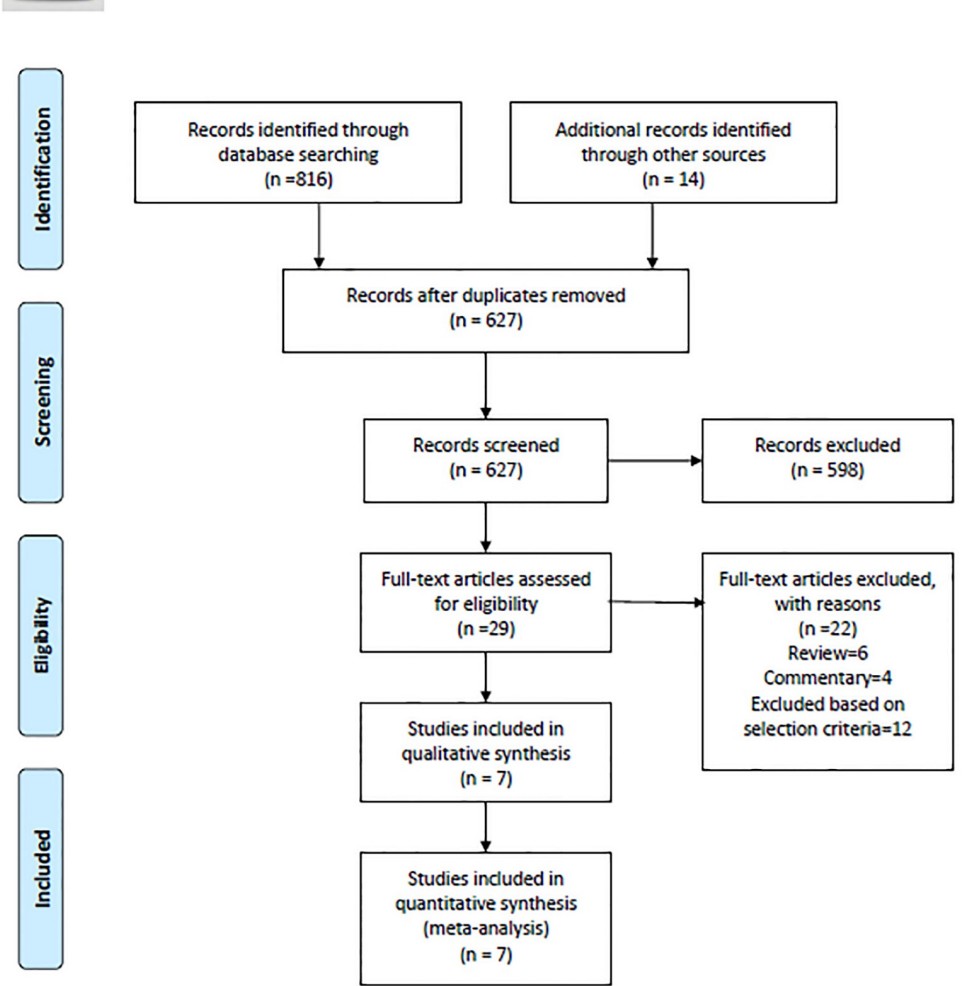

**Fig 1. Flow chart of study selection process after screening of electronic search.**

characteristics of the included studies are shown in Table 1. The trials by Gyamfi-Bannerman and Porto carried low ROB in most of the domains [10, 20] whereas those by Attawattanakul, Balci, Kasab, Mirzamoradi and Ontela, carried high to unclear ROB [21–25] (Fig 2).

## Primary outcomes

1. **Need for any respiratory support**: Five RCTs that included 3844 neonates (ANC: 1941, Control: 1903) reported this outcome [10, 20–22, 25]. Need for any respiratory support was significantly less in the ANC vs. control group neonates [ANC: 11% vs. Control: 16%]. Meta-analysis confirmed these findings [RR = 0.68(0.47–0.98), p = 0.04; Heterogeneity: $Chi^2$ = 8.84, $I^2$ = 55%; LOE: Moderate] (Fig 2). The number needed to treat (NNT) for preventing one case of respiratory support was 20 (Fig 3).

2. **Hypoglycaemia**: Four studies that reported this outcome included 3604 neonates (ANC: 1821, Control: 1783) [10, 20, 21, 25]. Incidence of hypoglycaemia was significantly high in

**Table 1. Characteristics of included randomised control trials*.**

| Studies | Type of steroid | Sample size | Main outcomes | Results |
|---|---|---|---|---|
| **Gyamifi-Bannerman 2016 USA** | Betamethasone 12mg/ 24 hrly x 2days | N:2827 ANC:1427 C:1400 | 1-Neonatal composite of treatment in the first 72 hours consists of one of the following a-Requirement of CPAP or HFNC for 2 hours, b- Supplemental oxygen with a Fio2≥ 0.30 for at least 4 hours, c- ECMO, or mechanical ventilation d- Stillbirth or neonatal death within 72 hours after delivery. 2- Hypoglycemia: Blood glucose levels 2.2 mmol per litre (<40 mg per decilitre) at any time. | 1-Neonatal composite of treatment: Reduced with ANC; RR: 0.80(0.66–0.97); P = 0.02 2-Hypoglycemia: High in ANC RR:1.60 (1.37–1.87); P<0.001 |
| **Porto 2011 Brazil** | Betamethasone 12mg/ 24 hrly x 2days | N:320 ANC:163 C:157 | Incidence of respiratory disorders: RDS or TTN, defined as presence of respiratory distress (tachypnoea, expiratory grunting, chest wall retractions, flaring of the nostrils, cyanosis, and increasing requirement of oxygen) for >2 hours after birth. | Respiratory disorders: No difference in two groups; Adjusted RR: 1.12 (0.74 to 1.70 |
| **Balci 2010 Turkey** | Betamethasone 12mg as single dose | N:100 ANC:50 C:50 | Requirement of resuscitation, RDS, Apgar score at 1 and 5 min | Requirement of resuscitation: ANC (14%) Vs control (32%); p = 0.032 RDS: ANC (4%) vs Control (16%); p = 0.046 Apgar Score: Better with ANC (p = 0.06 at 1 minutes and p<0.001 at 5 min) |
| **Kasab 2013 Egypt** | Dexamethasone 12mg IM/12hrly x 2 doses | N:200 ANC:100 C:100 | Requirement of resuscitation, RDS, Apgar score at 1 and 5 min | Need for resuscitation: ANC (14%) Vs control 32 (32%); p = 0.013 RDS: ANC (4%) vs Control (16%); p = 0.021 Apgar score: Better with ANC (1- and 5-minutes P<0.05). |
| **Mirzamoradi 2019 Iran** | Dexamethasone 12mg IM/24hrly x 2 doses | N:240 ANC:120 C:120 | Need for respiratory support by 72 h of life consisting of ≥1 of the following; CPAP or HFNC for ≥ 2 hours, RDS or need for ventilation. | Need for respiratory support: Significantly less in ANC group. Respiratory morbidity: Significantly less in ANC (16%) vs Control (50%); p<001. |
| **Ontela 2018 India** | Dexamethasone 6mg IM/12hrly x 4 doses | N:310 ANC:155 C:155 | Respiratory morbidities (TTN, RDS). TTN: Signs of respiratory distress at birth without significant X ray changes or with hyperinflation/ interlobar fluid that resolved in < 72 hours. RDS: Signs of respiratory distress at birth along with decreased air entry or diffuse granular infiltrates on X ray. | Respiratory morbidities: No difference between the two groups; RR 0.91 (0.7– 1.2); P = 0.32 |
| **Attawattanakul 2015 Thailand** | Dexamethasone 6mg IM/12hrly x 4 doses | N:194 ANC:96 C:98 | Incidence of respiratory distress: Grunting, flaring, tachypnoea > 60/min, retraction, and/or need for oxygen > 2 hours after birth. | Respiratory distress: Significantly less with ANC; RR 0.40, (0.17 to 0.94); p = 0.03 |

ANC: Antenatal corticosteroids, C: Control, CI: Confidence interval, CPAP: Continuous positive airway pressure, HFNC: High flow nasal cannula, ECMO: Extracorporeal membrane oxygenation OR: Odds ratio, RR: Relative risk, RDS- Respiratory distress syndrome, TTN- Transient tachypnoea of newborn

*Note- Neurodevelopment outcomes have not been reported by any of the included studies.

the ANC vs. control group (ANC: 20% vs. Control: 12.5%). Meta-analysis confirmed these findings [RR = 1.61(1.38–1.87), p<0.00001; Heterogeneity: $Chi^2$ = 0.40, $I^2$ = 0%; LOE: High] (Fig 3). The number needed for harm for hypoglycemia was 13 (Fig 4).

## Secondary outcomes

1. **Need for resuscitation at birth** [only in delivery room immediately after birth (not in NICU)]: The data for this outcome was available from 6 studies that included 3871 neonates (ANC: 1948, Control: 1923) [10, 21–25]. Need for resuscitation at birth was significantly less in the ANC vs. control group neonates [ANC: 10% vs. Control: 16.5%]. Meta-analysis

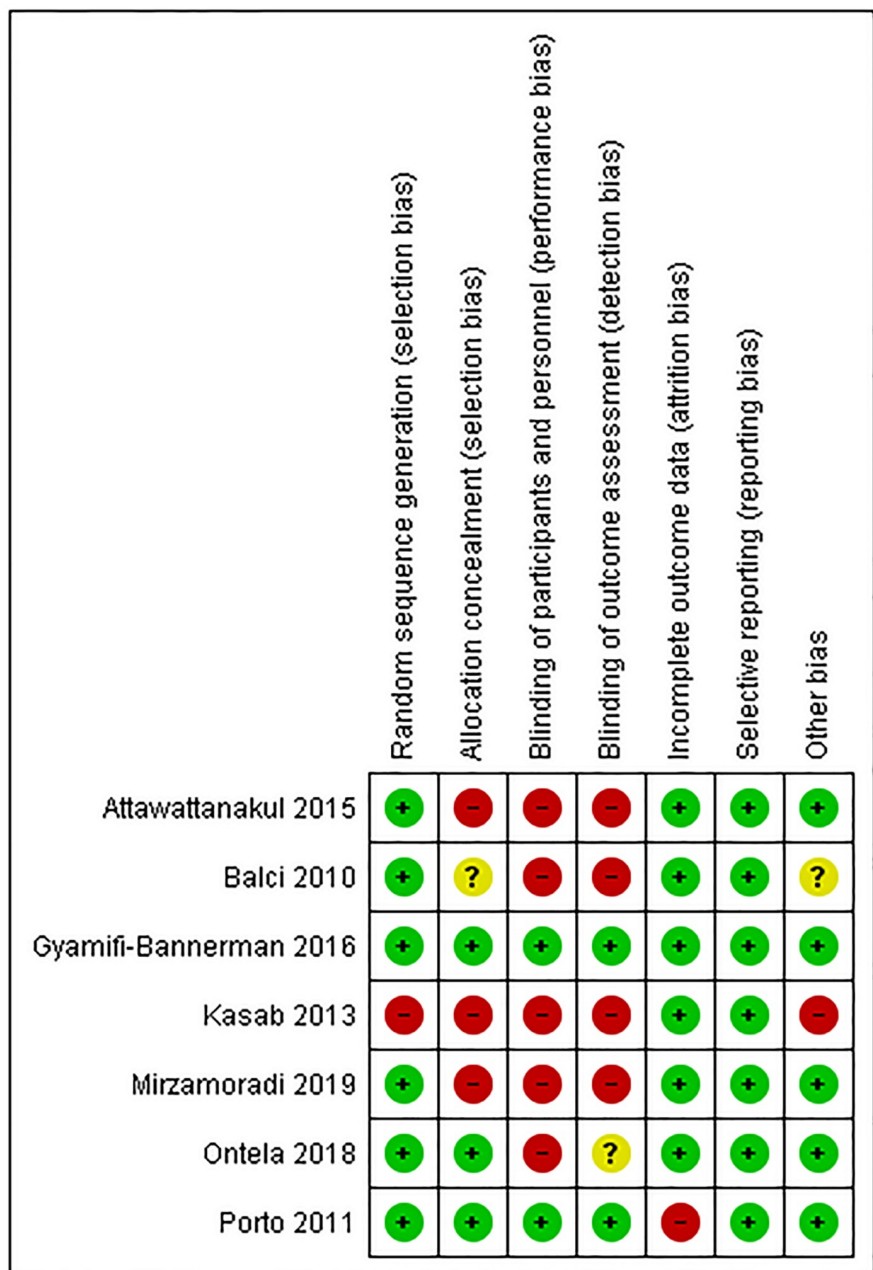

**Fig 2. Risk of bias summary.**

confirmed these findings [RR = 0.63(0.42–0.95), p = 0.03; Heterogeneity: Chi$^2$ = 8.73, I$^2$ = 43%; LOE: Low]. (S1 Fig) NNT for this outcome was 16.

2. **Admission to NICU**: Six studies that reported this outcome included 3944 neonates (ANC: 1991, Control: 1953) [10, 20–22, 24, 25]. There was no difference in admission to NICU in the ANC vs. control group neonates (ANC: 32% vs. Control: 38%). Meta-analysis confirmed these findings [RR = 0.84 (0.59–1.19), p = 0.32; Heterogeneity: Chi$^2$ = 18.44, I$^2$ = 73%; LOE: Low]. (S2 Fig)

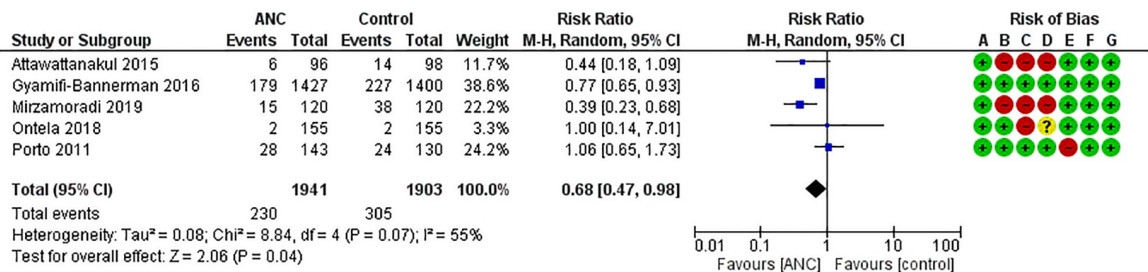

Fig 3. Effect of ANC on need for any respiratory support.

3. **TTN**: The incidence of TTN was reported in 5 RCTs including 3844 neonates (ANC: 1941, C: 1903) [10, 20–22, 25]. TTN was less in the ANC vs. control group (ANC: 10% vs. Control: 11.5 but was not statistically significant [RR = 0.90 (0.66–1.24) p = 0.53; $Chi^2$ = 7.57, $I^2$ = 47%; LOE: Low]. (S3 Fig)

4. **RDS**: All seven studies reported this outcome included 4143 neonates (ANC: 2090, C: 2053) [10, 20–25]. Incidence of RDS was less in the ANC vs. control group (ANC: 4% vs. Control:7%) but was not statistically significant [RR = 0.64 (0.35–1.17), p = 0.15; $Chi^2$ = 13.55, $I^2$ = 56%; LOE: Low]. (S4 Fig)

5. **Mortality**: Three studies that reported this outcome included 3200 neonates (ANC: 1821, Control: 1783) [10, 20, 24]. Mortality was similar in the ANC vs. control group neonates (ANC: 1% vs. Control: 1%). Meta-analysis confirmed these findings [RR = 0.94 (0.04–23.80), p = 0.97; Heterogeneity: $Chi^2$ = 2.27, $I^2$ = 56%; LOE: Very Low]. (S5 Fig)

6. **Need for mechanical ventilation**: The data for this outcome was available from 4 studies that included 3650 neonates (ANC: 1845, Control: 1805) [10, 20–22]. Need for mechanical

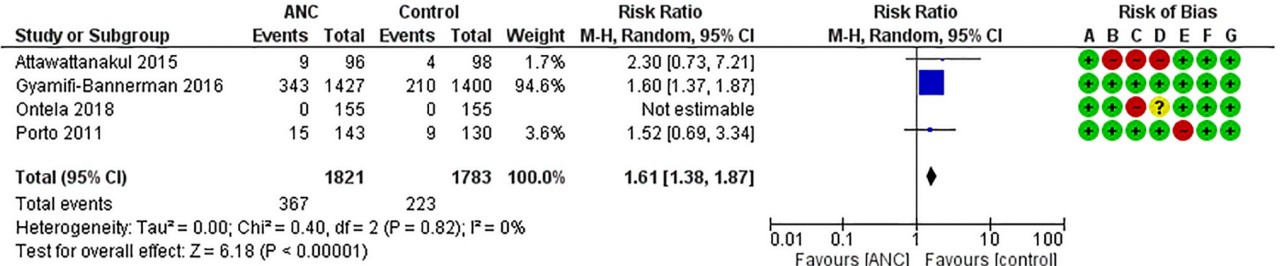

Fig 4. Effect of ANC on hypoglycemia.

ventilation was 21% in ANC vs. 27% in control group. Meta-analysis showed no difference between the two groups [RR = 0.78 (0.51–1.19), p = 0.25; Heterogeneity: $Chi^2$ = 1.09, $I^2$ = 0%; LOE: Moderate]. (S6 Fig)

7. **Need for Surfactant**: The data for this outcome was available from 3 studies that included 3340 neonates (ANC: 1690, Control: 1650) [10, 20, 22]. Need for surfactant was 1.5% in ANC vs. 3.2% in control group. Meta-analysis showed no difference between the two groups [RR = 0.45(0.11–1.84), p = 0.27; Heterogeneity: $Chi^2$ = 3.95, $I^2$ = 49%; LOE: Very Low]. (S7 Fig)

8. **Developmental follow up**: None of the included studies reported data on this outcome.

9. **Adverse effects**: Apart from neonatal hypoglycaemia none of the included studies reported any other adverse effects such as neonatal sepsis, and seizures.

**Sensitivity analysis.**   Results of the sensitivity analysis including only studies with low ROB (Gyamifi- Bannerman and Porto et al) showed no difference in any respiratory support between ANC vs. control group, hypoglycaemia was more, whereas need for resuscitation at birth and surfactant therapy were significantly less in ANC group [10, 20] (S1 Table).

**Subgroup analysis.**   This was not possible due to lack of stratified data based on gestational age, mode of delivery and maternal diabetes.

**Analysis using fixed effect model.**   The results of this comparative analysis showed that majority of the outcomes including both primary outcomes were similar with both models. However, the fixed effect model showed admission to NICU, need for surfactant, RDS and TTN were significantly less in ANC group. (S2 Table)

**Summary of findings table and publication bias.**   For the primary outcome of need for any respiratory support, the evidence was graded as moderate whereas it was deemed high for hypoglycaemia. For secondary outcomes the evidence was deemed as very low to moderate. (Table 2) Publication bias couldn't be ruled out due to a small number of trials [26].

## Quality of evidence GRADE

1. **High**: Risk of bias: not serious, Inconsistency: Not Serious, Indirectness: Not serious, Imprecision: Not serious, Other considerations: None

2. **Moderate**\*\*: Risk of bias: not serious, Inconsistency: Serious, Indirectness: Not serious, Imprecision: Not serious, Other considerations: None

3. **Moderate**$^{\$\$}$: Risk of bias: not serious, Inconsistency: Not Serious, Indirectness: Not serious, Imprecision: serious, Other considerations: None

4. **Low**[#]: Risk of bias: Serious, Inconsistency: Serious, Indirectness: Not serious, Imprecision: Not serious, Other considerations: None

5. **Low**\*: Risk of bias: Not Serious, Inconsistency: Serious, Indirectness: Not serious, Imprecision: serious, Other considerations: None

6. **Low**@@: Risk of bias: Serious, Inconsistency: Not Serious, Indirectness: Not serious, Imprecision: serious, Other considerations: None

7. **Very Low**@: Risk of bias: serious, Inconsistency: Serious, Indirectness: Non serious, Imprecision: Serious, Publication bias: Not serious

**Table 2. Summary of finding for pooled data as per GRADE guidelines.**

| Outcome | Absolute risk | | Relative effect | Number of participants | Quality of evidence |
|---|---|---|---|---|---|
| | | | RR (95% CI) | | GRADE[1] |
| **Effect of ANC on** | **Estimated risk with placebo** | **Corresponding risk with ANC** | | | |
| **Any respiratory support** | 160 per 1,000 | 109 per 1,000 (75 to 157) | RR 0.68 (0.47 to 0.98) | 3844 (5 RCTs) | ⊕⊕⊕◯ Moderate** |
| **Hypoglycemia** | 125per 1000 | 201 per 1000 (173 to234) | RR 1.61 (1.38 to 1.87) | 3604 (4 RCTs) | ⊕⊕⊕⊕ High |
| **Need for resus at birth** [only in delivery room immediately after birth (not in NICU)] | 165 per 1,000 | 104 per 1,000 (69 to 157) | RR 0.63 (0.42 to 0.95) | 3871 (6 RCTs) | ⊕⊕◯◯ Low# |
| **Admission to NICU** | 381 per 1,000 | 320 per 1,000 (225 to 453) | OR 0.84 (0.59 to 1.19) | 3944 (6 RCTs) | ⊕⊕◯◯ Low* |
| **TTN** | 114 per 1,000 | 102 per 1,000 (75 to 141) | RR 0.90 (0.66 to 1.24) | 3844 (5 RCTs) | ⊕⊕◯◯ Low@@ |
| **RDS** | 68 per 1,000 | 43 per 1,000 (24 to 79) | RR 0.64 (0.35 to 1.17) | 4143 (7 RCTs) | ⊕◯◯◯ Very Low@ |
| **Mortality** | 1 per 1,000 | 1 per 1,000 (0 to 30) | RR 0.94 (0.04 to 23.80) | 3200 (3 RCTs) | ⊕◯◯◯ Very Low$ |
| **Need for mechanical ventilation** | 27 per 1,000 | 21 per 1,000 (14 to 32) | RR 0.78 (0.51 to 1.19) | 3650 (4 RCTs) | ⊕⊕⊕◯ Moderate$$ |
| **Need for Surfactant** | 32 per 1000 | **15 per 1,000 (4 to 58)** | RR 0.45 (0.11 to 1.84) | 3340 (3RCT) | ⊕◯◯◯ Very Low$ |

**Abbreviations**: ANC: Antenatal corticosteroids, CI: Confidence interval, GRADE: Grading of Recommendations Assessment, Development and Evaluation, NICU: Neonatal intensive care unit, RDS: respiratory distress syndrome, RR: Relative risk, RCT: Randomised control trial, TTN: Transient tachypnoea of newborn

8. **Very Low$**: Risk of bias: not serious, Inconsistency: Serious, Indirectness: Non serious, Imprecision: Very serious, Other considerations: None

## Discussion

Our systematic review showed that exposure to ANC was beneficial in reducing the need for respiratory support but with the increased risk of hypoglycemia in neonates born at late preterm gestation. Exposure to ANC also reduced the need for resuscitation at birth. ANC had no impact on RDS, TTN, admission to NICU, need for mechanical ventilation, surfactant therapy and mortality. None of the included trials reported long term follow up data.

Respiratory morbidities in LPNs relate to developmental immaturity of the lungs. The mechanisms for benefits of ANC include enhanced alveolar differentiation with the induction of type 2 pneumocytes and activation of endothelial nitric oxide synthase [27, 28]. It is postulated that similar to term gestation, ANC exposure at late preterm gestation accelerates coordinated organ development sequence in response to endogenous rise in fetal glucocorticoids [29, 30]. Experimental studies show that the improvement in lung function after ANC exposure are due to an increase in the absorption of fetal lung fluid, thinning of alveolar septae, and synthesis of surfactant proteins and phospholipids [30, 31].

The benefits of ANC need to be considered in the context of their potential adverse effects. ANC exposure is associated with reduced brain mass, delayed myelination, decreased maturation of peripheral nerves, increased impairment of hypothalamopituitory axis and impaired programmed apoptosis in animal studies [30, 32–34]. A follow-up study of participants from ASTECS trial reported two-fold increase in teacher-reported low academic ability in children

(age 8–15 years) exposed to betamethasone as term infants born by an elective caesarean section. There were no significant differences in general health, behaviour and academic achievements between exposed vs. unexposed groups. However, only 51% response rate diminishes the validity of these finding [35].

A recent population-based study from Finland using nationwide registries of all (term and preterm) singleton live births found that ANC exposure was associated with a significant increase [adjusted hazard ratio (aHR): 1.33; 95% CI (1.26–1.41] in mental and behavioural disorders in children at 5.8 (interquartile-range, 3.1–8.7) years. The incidence of these disorders was high in term neonates exposed to ANC vs. controls [aHR: 1.38; (95% CI: 1.21–1.58)]. The incidence of these adverse outcomes was higher (14.59% vs 10.71%) but statistically non-significant [aHR: 1.00; (95% CI: 0.92–1.09)] in preterm neonates [36].

It is important to consider the implications of our results for clinical practice. The NNT to prevent one case of any respiratory support and need for resuscitation at birth was 20 and 16 respectively. In comparison, the NNT for harm was 13 for neonatal hypoglycaemia. LPNs are at high risk of hypoglycaemia due to poor substrate and underdeveloped compensatory response. The increased risk of neonatal hypoglycaemia might be due to transient hyperinsulinemia following maternal hyperglycaemia in response to ANC [37].

The definition of neonatal hypoglycaemia varied in the RCTs included in our review. The ALPS study defined it as blood glucose <2.2mmol/L [10]. This raises the possibility that many neonates with hypoglycemia may have been missed considering the widely accepted definition of hypoglycaemia is blood glucose <2.6 mmol/L. Hypoglycaemia is an independent predictor of poor neurodevelopmental outcomes in neonates. A large prospective cohort study from Sweden (n = 101,060) found that, in infants with early moderate hypoglycemia (<6 hours after birth, blood glucose<2.2mmol/L), the risk of any adverse neurological or neurodevelopmental outcome and cognitive developmental delay was increased by two [OR 1.94 (1.30–2.89)] and three [OR 3.17 (1.35–7.43)] fold respectively compared to normoglycaemic infants [38].

A recent systematic review of non-RCTs (N = 1395) showed that neonatal hypoglycaemia was associated with visual-motor impairment (n = 508; OR = 3.46, 95% CI = 1.13–10.57) and executive dysfunction (n = 463; OR = 2.50, 95% CI = 1.20–5.22) in early childhood. In mid-childhood, the odds of neurodevelopmental impairment (n = 54; OR = 3.62, 95% CI = 1.05–12.42), low literacy (n = 1,395; OR = 2.04, 95% CI = 1.20–3.47) and numeracy (n = 1,395; OR = 2.04, 95% CI = 1.21–3.44) were significantly higher [39].

Inability to predict spontaneous preterm birth means inevitable unwarranted exposure to ANC in a significant number of late preterm pregnancies, which do not result in late preterm delivery. The ALP study reported that 16% of pregnancies with ANC exposure went on to deliver at term gestation [28]. Overall, potential long-term neurodevelopmental adverse effects of ANC must receive due attention considering that none of the included trials till date have reported such data.

The limitations of using Apgar scores as a primary outcome need to be discussed. Apgar score at 1 minute does not correlate with mortality and long-term neurodevelopmental outcomes [40]. Apgar scores can be low in otherwise well preterm neonates with no evidence of perinatal asphyxia [40–42]. Moreover, being a continuous measure, it is much more likely that differences would be noted regardless of their clinical importance.

The strengths of our review include its robust methodology, inclusions of only RCTs, large sample size and use of GRADE guidelines for summarizing the level of evidence. We conducted sensitivity analysis excluding the studies with high ROB. Compared to the previous systematic review we have provided data from 4 more RCTs (n~1000), all from developing countries. The provision of the NNT for both, benefit and harm, is important for guiding

research and clinical practice. The limitations of our review include the fact that the pooled results are still influenced by the large ALPS trial [10]. The included RCTs differ in the definitions of various outcomes, type of steroids, and their dosage. Furthermore, analysis stratified by gestation, mode of delivery or maternal diabetic status was not possible.

## Conclusions

Our systematic review showed that exposure to ANC was beneficial in reducing the need for respiratory support but with an increased risk of hypoglycemia in neonates born at late preterm gestation. Exposure to ANC also reduced the need for resuscitation at birth.

In summary, moderate to low-quality evidence indicates that ANC exposure reduced the need for respiratory support and increased the risk of neonatal hypoglycaemia in LPNs. ANC reduced need for resuscitation at birth in LPNs. The increased risk of neonatal hypoglycaemia is a serious concern. Pragmatic and adequately powered multicentre RCTs with long-term follow up assessing neonatal neurodevelopmental outcomes are needed to assess the efficacy and safety of ANC. Stratification by gestation, mode of delivery, maternal diabetes and other risk factors for respiratory distress is desirable in such trials. Pending results of such trials rigorous monitoring, treatment, and follow up of LPNs exposed to ANC is critical, more so in the context of hypoglycemia.

## Supporting information

**S1 Fig. Effect of ANC on need for resuscitation at birth.**
(TIF)

**S2 Fig. Effect of ANC on admission to nursery.**
(TIF)

**S3 Fig. Effect of ANC on TTN.**
(TIF)

**S4 Fig. Effect of ANC on RDS.**
(TIF)

**S5 Fig. Effect of ANC on mortality.**
(TIF)

**S6 Fig. Effect of ANC on mechanical ventilation.**
(TIF)

**S7 Fig. Effect of ANC on need for surfactant.**
(TIF)

**S1 Table. Results of sensitivity analysis based on ROB.**
(DOCX)

**S2 Table. Results of analysis by fixed and random effects model.**
(DOCX)

**S3 Table. Compliance of PRISMA guidelines.**
(DOC)

**S1 File. Search strategy.**
(DOCX)

## Acknowledgments

We thank Ms Rhonda Mayberry (RM), Librarian, Fiona Stanley hospital, Perth, WA, for conducting an updated literature search.

**Submission declaration**: All authors declare that the work submitted has not been published previously, that it is not under consideration for publication elsewhere, that its publication is approved by all authors and tacitly or explicitly by the responsible authorities where the work was carried out, and that, if accepted, it will not be published elsewhere including electronically in the same form, in English or in any other language, without the written consent of the copyright-holder.

## Author Contributions

**Conceptualization:** Mangesh Deshmukh, Sanjay Patole.

**Data curation:** Mangesh Deshmukh.

**Formal analysis:** Mangesh Deshmukh.

**Methodology:** Mangesh Deshmukh, Sanjay Patole.

**Software:** Mangesh Deshmukh.

**Supervision:** Sanjay Patole.

**Validation:** Mangesh Deshmukh.

**Visualization:** Mangesh Deshmukh.

**Writing – original draft:** Mangesh Deshmukh, Sanjay Patole.

**Writing – review & editing:** Mangesh Deshmukh, Sanjay Patole.

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
