## [Decision Letter · Decision Letter 0]

14 Dec 2020

PONE-D-20-35297

Antenatal Corticosteroids for Impending Late Preterm (34-36+6 Weeks) Deliveries – A systematic review of RCTs

PLOS ONE

Dear Dr. Mangesh Deshmukh,

Thank you for submitting your manuscript to PLOS ONE. After careful consideration, we feel that it has merit but does not fully meet PLOS ONE’s publication criteria as it currently stands. Therefore, we invite you to submit a revised version of the manuscript that addresses the points raised during the review process.

We look forward to receiving your revised manuscript.

Kind regards,

Georg M. Schmölzer

Academic Editor

PLOS ONE

Journal Requirements:

2. Please ensure you have included details of data extraction methods in the manuscript methods. We would expect to see reporting of the specific information extracted from the manuscripts.

3. We note that your systematic review includes a meta-analysis. Please update the title to reflect this, e.g. update to "systematic review and meta-analysis".

4.Thank you for submitting the above manuscript to PLOS ONE. During our internal evaluation of the manuscript, we found significant text overlap between your submission and the following previously published works, some of which you are an author:

https://journals.plos.org/plosone/article?id=10.1371%2Fjournal.pone.0176090

https://academic.oup.com/tropej/article/64/6/531/4819249

https://www.tandfonline.com/doi/abs/10.1080/14767058.2018.1554051?af=R&journalCode=ijmf20

https://www.nejm.org/doi/10.1056/NEJMoa1516783?page=9&sort=newest

https://www.bmj.com/content/342/bmj.d1696?ijkey=88b473f74af7363bb70afc21e981af2f41853d6c&keytype2=tf_ipsecsha

Please revise the manuscript to rephrase the duplicated text, cite your sources, and provide details as to how the current manuscript advances on previous work. Please note that further consideration is dependent on the submission of a manuscript that addresses these concerns about the overlap in text with published work.

Reviewers' comments:

Reviewer's Responses to Questions

**Comments to the Author**

1. Is the manuscript technically sound, and do the data support the conclusions?

Reviewer #1: Yes

2. Has the statistical analysis been performed appropriately and rigorously? 

Reviewer #1: Yes

3. Have the authors made all data underlying the findings in their manuscript fully available?

Reviewer #1: Yes

4. Is the manuscript presented in an intelligible fashion and written in standard English?

Reviewer #1: Yes

5. Review Comments to the Author

Reviewer #1: Thank you for the opportunity to review this manuscript, which tackles the important topic of ACS use in late preterm infants.

An appropriately rigorous methodology was generally used, following guidelines.

The manuscript is generally well written and clear with a logical flow and a balanced approach.

A) The major comment I have is that it does not appear that the protocol was registered with Prospero or another database.

This should be stated outright in the method section. This is unfortunate as it is that’s impossible to determine what were a priori defined results versus post hoc. However, these should still be labelled a priori versus post hoc by the authors.

Aii)

Apgar score does not appear to be listed in the outcomes in the Methods. Thus it should be labelled post hoc, and any others that were not planned

B) The second major issue is: The authors have listed under the outcome “need for resuscitation at birth“ “positive pressure ventilation, CPAP, facial oxygen“.

Bi) These appear to be respiratory support not resus.

Bii) Indeed some of the features overlap with the authors’ primary outcome “need for any respiratory support (mechanical ventilation, CPAP, high flow) or oxygen“.

Thus the results in these 2 outcomes (need for resuscitation at birth“ “positive pressure ventilation, CPAP, facial oxygen“ and “need for any respiratory support (mechanical ventilation, CPAP, high flow) or oxygen“), overlap ?completely or very significantly, essentially meaning the same outcomes were double counted and is inappropriate. A different secondary outcome without overlap should be created.

C) In table 1, the authors list “results” however this is only a partial list of the results in the trials. Please change the heading to be much more descriptive – which results did you choose to list in table one?

D) In table 1, some of the results that are presented our odds ratios which seems very odd if this was actually an RCT.

Please verify

E) Please incorporate the GRADE into your results int the body of the manuscript e.g. In lines 195 to 241, Just as you did in the abstract

F) given that the authors to a priority defined outcomes including both respiration and hypoglycemia, the summary statements of the results in the abstract as well as in the body of the manuscript should reflect this with both outcomes in the first sentence, and not hypoglycaemia relegated to the second sentence. This better reflects there a priority defined primary outcomes, but also avoids misinterpretation of results should people read only the first sentence– Which can occur

G) outcomes which had no data should still be included in Tables, has this emphasizes the need for data for those August and also remind Reader’s of your initial intention to try to inform the subject in this area hence for example developmental outcomes even if there is no data should be added to all of the tables

H) The use of a primary outcome in many of the included RCTs of Apgar score at one in five minutes is problematic, and this should mention this in the Discussion.

• Apgar score at 1 minute reflects resuscitation/stabilization and does not correlate with long term outcomes.

• Apgar score in preterm infants has not been validated to my knowledge

• Because it is a continuous measure it is much more likely that differences would be noted, Regardless of clinical importance

Minor revisions:

1) Please define “facial oxygen“. Since all oxygen is applied somehow to the face, I’m not sure that this description is clear – does this include both high and low flow nasal prongs?

2) In supplemental table one please bold only the statistically significant results

Please include the actual search strategies used in each database in your supplement

3) in line 27 suggest replacing the word neonate with infants to avoid confusion that (Neonatal typically defined within the first 28 days of life and the authors appropriately had a broader scope for their SR)

4 line 45 suggest revision of the statement that large trials are needed – as many would argue that ALPS was a large trial, I believe what the authors mean is that large trials with adequate follow up for Neurodevelopmental outcomes are needed

5) even in the abstract I think it would be worth stressing that of the included Trials, only a single one was in HIC the others were not

Well written manuscript that adds importantly to the literature. Congratulations

6. PLOS authors have the option to publish the peer review history of their article (what does this mean?). If published, this will include your full peer review and any attached files.

Reviewer #1: **Yes: **Sarah D McDonald

---

## [Author Response · Author response to Decision Letter 0]

17 Feb 2021

Dear Editor

We appreciate the opportunity to revise our manuscript based on the reviewer’s comments. Please find the point-by-point response as follows. Hope to hear from you soon.

Dr Mangesh Deshmukh

Dr Sanjay Patole

Editorial Comments:

Response: Done

2. Please ensure you have included details of data extraction methods in the manuscript methods. We would expect to see reporting of the specific information extracted from the manuscripts.

Response: We have updated the manuscript in data extraction section incorporating the suggestions. 

3. We note that your systematic review includes a meta-analysis. Please update the title to reflect this, e.g. update to "systematic review and meta-analysis".

Response: Done

4.Thank you for submitting the above manuscript to PLOS ONE. During our internal evaluation of the manuscript, we found significant text overlap between your submission and the following previously published works, some of which you are an author:

Response: Apologies for the overlap in Table-1 (Characteristics of included randomised control trials) particularly in the 'Primary Outcome' row. We have now updated the table by rephrasing the terms as much as possible. We also updated one more sentence in Discussion section as follows (Page 17; Lines 313-15)

Experimental studies show that the improvement in lung function after ANC exposure are due to an increase in the absorption of fetal lung fluid, thinning of alveolar septae, and synthesis of surfactant proteins and phospholipids.[30, 31]

Reviewer’s comments

Ai) The major comment I have is that it does not appear that the protocol was registered with Prospero or another database. This should be stated outright in the method section. This is unfortunate as it is that’s impossible to determine what were a priori defined results versus post hoc. However, these should still be labelled a priori versus post hoc by the authors.

Response: We have not registered the protocol with PROSPERO or any other database.

As advised, we have mentioned this in the Material and methods section Page 6, line102-03

Aii) Apgar score does not appear to be listed in the outcomes in the Methods. Thus, it should be labelled post hoc, and any others that were not planned

Response: Apgar score is not an outcome we studied and hence not reported in our results.

Apgar scores were the outcome reported by the investigators of included trials. Hence, we have included them in Table 1(Characteristics of included randomised control trials), which gives characteristics and key findings of included trials.

B) The second major issue is: The authors have listed under the outcome “need for resuscitation at birth”, “positive pressure ventilation”, “CPAP”, “facial oxygen”.

Bi) These appear to be respiratory support not resus.

Bii) Indeed some of the features overlap with the authors’ primary outcome “need for any respiratory support (mechanical ventilation, CPAP, high flow) or oxygen“.

Thus the results in these 2 outcomes (need for resuscitation at birth“ “positive pressure ventilation, CPAP, facial oxygen“ and “need for any respiratory support (mechanical ventilation, CPAP, high flow) or oxygen“), overlap completely or very significantly, essentially meaning the same outcomes were double counted and is inappropriate. A different secondary outcome without overlap should be created.

Response: Apologies for the confusion. The outcome defined as “Need for resuscitation at birth” was meant to include, requirement of any intervention including positive pressure ventilation, CPAP, facial oxygen only in the delivery room immediately after birth.

Our primary outcome “Need for any respiratory support” (in NICU) is different from need for resuscitation in the delivery room as defined above. There is no overlap. 

Total 5/6 of the included RCTs reported outcome of “need for resuscitation at birth”, we got the clarification from the remaining RCT where the outcome definition was not clear (Ontela et al) confirming no overlap of the data between the two outcomes.

We have modified the sentence as below to clarify this issue in “Materials and method” section. Page 6, Line-112-14.

Need for resuscitation at birth: Requirement of any intervention including positive pressure ventilation, CPAP, facial oxygen (defined as free flow of oxygen near the nostrils by a catheter or mask) only in the delivery room immediately after birth.

C) In table 1, the authors list “results” however this is only a partial list of the results in the trials. Please change the heading to be much more descriptive – which results did you choose to list in table one?

Response: We have included primary outcomes from original studies and their secondary outcomes relating to adverse effects of ANC (e.g. hypoglycaemia). 

D) In table 1, some of the results that are presented our odds ratios which seems very odd if this was actually an RCT. Please verify

Response: The published data indeed reports OR instead of RR despite the study being a RCT. We assume that the journal accepted ORs considering they are close to RRs when the event rate is low. However, to avoid this confusion now we have given percentage of each outcome. Table 1(Characteristics of included randomised control trials)

E) Please incorporate the GRADE into your results in the body of the manuscript e.g. In lines 195 to 241, Just as you did in the abstract.

Response: We have incorporated the GRADE into results in the body of the manuscript as suggested. We have also made a small change in Grading the evidence and summary of findings section (Page 8, lines 173) instead of “quality” to “level of evidence (LOE)”

We used the Grading of Recommendations Assessment, Development and Evaluation (GRADE) for assessment and Grade pro guidelines development tool to create the summary of findings table for reporting the level of evidence (LOE). [18, 19]

F) Given that the authors to a priority defined outcome including both respiration and hypoglycemia, the summary statements of the results in the abstract as well as in the body of the manuscript should reflect this with both outcomes in the first sentence, and not hypoglycaemia relegated to the second sentence. This better reflects there a priority defined primary outcomes, but also avoids misinterpretation of results should people read only the first sentence– Which can occur

Response: We have modified the first few lines of “Discussion” (Page 17, Lines 303-305) and “Conclusions” (Page 20, Lines 378-81) as follows 

• Our systematic review showed that exposure to ANC was beneficial in reducing the need for respiratory support but increased the risk of hypoglycaemia in neonates born at late preterm gestation. Exposure to ANC also reduced the need for resuscitation at birth.

• In summary, moderate to low quality evidence indicates that ANC exposure reduced the need for respiratory support and increased the risk of neonatal hypoglycaemia in LPNs. ANC reduced need for resuscitation at birth in LPNs. The increased risk of neonatal hypoglycaemia is a serious concern.

G) Outcomes which had no data should still be included in Tables, has this emphasizes the need for data for those August and also remind readers of your initial intention to try to inform the subject in this area hence for example developmental outcomes even if there is no data should be added to all of the tables.

Response: Assessment of long-term neurodevelopmental follow up was not the aim of our systematic review. We focused only the short-term outcomes in the included studies. Furthermore, none of the included trails reported such data. We have added a footnote to Table 1(Characteristics of included randomised control trials) as follows: 

“Neurodevelopment outcomes have not been reported by any of the included studies.”

H) The use of a primary outcome in many of the included RCTs of Apgar score at one in five minutes is problematic, and this should mention this in the Discussion.

Apgar score at 1 minute reflects resuscitation/stabilization and does not correlate with long term outcomes. Apgar score in preterm infants has not been validated to my knowledge. Because it is a continuous measure it is much more likely that differences would be noted, Regardless of clinical importance.

Response: Thank you for your comments, we have added following sentences in the Discussion section (Page 19, Lines 360-364)

The concerns about using Apgar scores as a primary outcome need to be discussed. Apgar score at 1 minute does not correlate with mortality and long-term neurodevelopmental outcomes. [40] Apgar scores can be low in otherwise well preterm neonates with no evidence of perinatal asphyxia. [40-42] Moreover, being a continuous measure, it is much more likely that differences would be noted regardless of their clinical importance.

Minor revisions:

1) Please define “facial oxygen”. Since all oxygen is applied somehow to the face, I’m not sure that this description is clear – does this include both high and low flow nasal prongs?

Response: Facial oxygen was defined as free flow of oxygen near the nostrils by a catheter or mask. This is a common practise especially in resource limited set up. We have provided this definition in material and methods section (Page 6, Lines 113-14).

2) In supplemental table one please bold only the statistically significant results

Response: Done

Please include the actual search strategies used in each database in your supplement

Response: We have now included search strategy for Medline via Ovid including Epub ahead of print, Embase, Web of science. We have also fixed the issue with formatting. Details are in S1 file supporting information– Search strategy

3) in line 27 suggest replacing the word neonate with infants to avoid confusion that (Neonatal typically defined within the first 28 days of life and the authors appropriately had a broader scope for their SR)

Response: Done.

4) line 45 suggest revision of the statement that large trials are needed – as many would argue that ALPS was a large trial, I believe what the authors mean is that large trials with adequate follow up for Neurodevelopmental outcomes are needed

Response: Done. (Page 2, 3, Lines 46-48)

5) even in the abstract I think it would be worth stressing that of the included Trials, only a single one was in HIC the others were not

Response: Done. (Page 2, Lines 37-38)

---

## [Decision Letter · Decision Letter 1]

3 Mar 2021

PONE-D-20-35297R1

Antenatal Corticosteroids for Impending Late Preterm (34-36+6 Weeks) Deliveries – A Systematic review and Meta-analysis of RCTs

PLOS ONE

Dear Dr. Mangesh Deshmukh,

Thank you for submitting your manuscript to PLOS ONE. After careful consideration, we feel that it has merit but does not fully meet PLOS ONE’s publication criteria as it currently stands. Therefore, we invite you to submit a revised version of the manuscript that addresses the points raised during the review process.

We look forward to receiving your revised manuscript.

Kind regards,

Georg M. Schmölzer

Academic Editor

PLOS ONE

Journal Requirements:

Reviewers' comments:

Reviewer's Responses to Questions

**Comments to the Author**

1. If the authors have adequately addressed your comments raised in a previous round of review and you feel that this manuscript is now acceptable for publication, you may indicate that here to bypass the “Comments to the Author” section, enter your conflict of interest statement in the “Confidential to Editor” section, and submit your "Accept" recommendation.

Reviewer #1: (No Response)

2. Is the manuscript technically sound, and do the data support the conclusions?

Reviewer #1: Yes

3. Has the statistical analysis been performed appropriately and rigorously? 

Reviewer #1: Yes

4. Have the authors made all data underlying the findings in their manuscript fully available?

Reviewer #1: Yes

5. Is the manuscript presented in an intelligible fashion and written in standard English?

Reviewer #1: Yes

6. Review Comments to the Author

Reviewer #1: Thank you for the opportunity to re-review this manuscript. I appreciate the authors responses.

A) The authors responded clearly in their response to the comments on page 2 however for clarity in the 1)abstract, 2)manuscript and in the 3)tables, please add to “need for resuscitation at birth” adding “only in the delivery room immediately after birth (Not in NICU).

B) Please also put back(I know it had been removed witt TC, but I did not request this removal and it is important) in your conclusion of the abstract that large definitive trials are required to assess benefits and risks of ANC in this population

C) Please note that in track changes line 384 states “the concerns about using Apgar scores as a primary outcome need to be discussed“ please remove the sentence as this was my direction to you

7. PLOS authors have the option to publish the peer review history of their article (what does this mean?). If published, this will include your full peer review and any attached files.

Reviewer #1: **Yes: **Professor Sarah D McDonald

---

## [Author Response · Author response to Decision Letter 1]

3 Mar 2021

Dear Editor

We appreciate the opportunity to revise our manuscript based on the reviewer’s comments. Please find the point-by-point response as follows. Hope to hear from you soon.

Dr Mangesh Deshmukh

Dr Sanjay Patole

Reviewer #1: 

A) The authors responded clearly in their response to the comments on page 2 however for clarity in the 1)abstract, 2)manuscript and in the 3)tables, please add to “need for resuscitation at birth” adding “only in the delivery room immediately after birth (Not in NICU).

Response: Done; 

Abstract: Page 2, Line 35,36; 

Manuscript: Material and methods: Page 6, line 115-16, Results: Page 12, lines 226-27,

Table 2

B) Please also put back(I know it had been removed witt TC, but I did not request this removal and it is important) in your conclusion of the abstract that large definitive trials are required to assess benefits and risks of ANC in this population

Response: Thanks for the comment. The above sentence is already there in the abstract (Page 2,3, Line 47-49)

Large definitive trials with adequate follow up for neurodevelopmental outcomes are required to assess benefits and risks of ANC in this population.

C) Please note that in track changes line 384 states “the concerns about using Apgar scores as a primary outcome need to be discussed“ please remove the sentence as this was my direction to you

Response: We used the sentence as a linking sentence to introduce discussion on APGAR score.

We have modified the sentence as follows, in Discussion section Page 19, line 364.

The limitations of using Apgar scores as a primary outcome need to be discussed.

---

## [Editor Report · Decision Letter 2]

5 Mar 2021

Antenatal Corticosteroids for Impending Late Preterm (34-36+6 Weeks) Deliveries – A Systematic review and Meta-analysis of RCTs

PONE-D-20-35297R2

Dear Dr. Mangesh Deshmukh,

We’re pleased to inform you that your manuscript has been judged scientifically suitable for publication and will be formally accepted for publication once it meets all outstanding technical requirements.

Kind regards,

Georg M. Schmölzer

Academic Editor

PLOS ONE
---

## [Editor Report · Acceptance letter]

11 Mar 2021

PONE-D-20-35297R2 

Antenatal Corticosteroids for Impending Late Preterm (34-36+6 Weeks) Deliveries – A Systematic review and Meta-analysis of RCTs 

Dear Dr. Deshmukh:

I'm pleased to inform you that your manuscript has been deemed suitable for publication in PLOS ONE. Congratulations! Your manuscript is now with our production department. 

Kind regards, 

on behalf of

Dr. Georg M. Schmölzer 

Academic Editor

PLOS ONE